**The role of $H_2SO_4$-$NH_3$ anion clusters in ion-induced aerosol nucleation**
**mechanisms in the boreal forest**
Chao Yan[1], Lubna Dada[1], Clémence Rose[1], Tuija Jokinen[1], Wei Nie[1,2], Siegfried
Schobesberger[1,3], Heikki Junninen[1,4], Katrianne Lehtipalo[1], Nina Sarnela[1], Ulla Makkonen[5],
Olga Garmash[1], Yonghong Wang[1], Qiaozhi Zha[1], Pauli Paasonen[1], Federico Bianchi[1], Mikko
Sipilä[1], Mikael Ehn[1], Tuukka Petäjä[1,2], Veli-Matti Kerminen[1], Douglas R. Worsnop[1,6], Markku
Kulmala[1,2,7]
[1] Institute for Atmospheric and Earth System Research / Physics, Faculty of Science, University of
Helsinki, P.O. Box 64, FI-00014, Helsinki, Finland
[2] Joint International Research Laboratory of Atmospheric and Earth System Sciences, School of
Atmospheric Sciences, Nanjing University, Nanjing, 210046, P.R. China
[3] Department of Applied Physics, University of Eastern Finland, 70211 Kuopio, Finland
[4] Institute of Physics, University of Tartu, Ülikooli 18, EE-50090 Tartu, Estoni
[5] Finnish Meteorological Institute, 00560 Helsinki, Finland.
[6] Aerodyne Research, Inc., Billerica, MA 01821, USA
[7] Aerosol and Haze Laboratory, Beijing Advanced Innovation Center for Soft Matter Science and
Engineering, Beijing University of Chemical Technology, Beijing, 100029, P.R. China
Correspondence to: Chao Yan (chao.yan@helsinki.fi)
**Abstract**
New particle formation (NPF) provides a large source of atmospheric aerosols, which affect
the climate and human health. Ion-induce nucleation (IIN) has been discovered as an important
pathway of forming particles within recent chamber studies, however, atmospheric
investigation remains incomplete. For this study, we investigated the air anion compositions in
the boreal forest in Southern Finland for 3 consecutive springs, with a special focus on $H_2SO_4$-
$NH_3$ anion clusters. We found that the ratio between the concentrations of highly oxygenated
organic molecules (HOMs) and $H_2SO_4$ controlled the appearance of $H_2SO_4$-$NH_3$ clusters (3<
#S < 13): All such clusters were observed when [HOM]/[$H_2SO_4$] was smaller than 30. The
number of $H_2SO_4$ molecules in the largest observable cluster correlated with the probability of
ion-induced nucleation (IIN) occurrence, which reached almost 100 % when the largest
observable cluster contained 6 or more $H_2SO_4$ molecules. During selected cases when the time
evolution of $H_2SO_4$-$NH_3$ clusters could be tracked, the calculated ion growth rates exhibited a
good agreement across measurement methods and cluster (particle) sizes. In these cases,
$H_2SO_4$-$NH_3$ clusters alone could explain ion growth up to 3 nm (mobility diameter). IIN events
also occurred in the absence of $H_2SO_4$-$NH_3$, implying that also other NPF mechanisms prevail
at this site, most likely involving HOMs. It seems that $H_2SO_4$ and HOMs both affect the
occurrence of an IIN event, but their ratio ([HOMs]/[$H_2SO_4$]) defines the primary mechanism
of the event. Since that ratio is strongly influenced by solar radiation and temperature, IIN
mechanism ought to vary depending on conditions and seasons.

**1  Introduction**
Atmospheric aerosol particles are known to influence human health and the climate (Heal et
al., 2012; Stocker et al., 2013). New particle formation (NPF) from gas-phase precursors
contributes to a major fraction of the global cloud condensation nuclei population (Merikanto
et al., 2009; Kerminen et al., 2012; Dunne et al., 2016; Gordon et al., 2017), and provides an
important source of particulate air pollutants in many urban environments (Guo et al., 2014).
Although NPF is an abundant phenomenon and has been observed in different places around
the globe within the boundary layer (Kulmala et al., 2004), the detailed mechanisms at each
location may differ and are still largely unknown. Experiments done in the CLOUD chamber
(Cosmic Leaving Outside Droplets) at CERN explored different NPF mechanisms on
molecular level, including sulfuric acid ($H_2SO_4$) and ammonia ($NH_3$) nucleation (Kirkby et al.,
2011), $H_2SO_4$ and dimethylamine (DMA) nucleation (Almeida et al., 2013), and pure biogenic
nucleation (Kirkby et al., 2016) from highly oxygenated organic molecules (HOMs) (Ehn et
al., 2014). While chamber experiments can mimic some properties of ambient observations
(Schobesberger et al., 2013), it is still ambiguous to what extent these chamber findings can be
applied to understand NPF in the more complex atmosphere, mostly due to the challenges in
atmospheric measurements and characterization of the nucleating species.
In the aforementioned chamber studies, ions have been shown to play a crucial role in
enhancing new particle formation, which is known as ion-induced nucleation (IIN). The
importance of IIN varies significantly depending on the temperature as we as the concentration
and composition of the ion species. For instance, big $H_2SO_4$ ion clusters were not found in the
sulfur-rich airmass from Atlanta, suggesting the minor role of IIN (Eisele et al., 2006). Similar
conclusions were drawn based on the observations in Boulder (Iida et al., 2006) and Hyytiälä
(e.g., Manninen et al., 2010), although the suggested importance of IIN in cold environment,
such as upper troposphere, cannot be excluded (Lovejoy et al., 2004 Kurten et al., 2016).
Recent the CLOUD experiments have revealed that the importance of IIN can be negligible in
the $H_2SO_4$-DMA system (Almeida et al., 2013), moderate in the $H_2SO_4$-$NH_3$ system (Kirkby
et al., 2011) and dominating in the pure HOMs system (Kirkby et al., 2016). However, it is
also important to note that the ion-pair concentration in Hyytiälä is lower than in the CLOUD
chamber, which partly explains the its smaller contribution of IIN (Wagner et al., 2017).

The recently developed atmospheric-pressure-interface time-of-flight mass spectrometer (APi-TOF) (Junninen et al., 2010) has been used for measuring ion composition at the SMEAR II station in Hyytiälä since 2009. Ehn et al., (2010) have first shown that the negative ion population varied significantly, with $H_2SO_4$ clusters dominating during the day and $HOM-NO_3^-$ clusters during the night. This variation was further studied by Bianchi et al., (2017), who grouped HOM-containing ions by separating the HOMs into non-nitrate- and nitrate-containing species as well as into ion adducts with $HSO_4^-$ or $NO_3^-$. In the night time, HOMs may form negatively charged clusters containing up to 40 carbons (Bianchi et al., 2017; Frege et al., 2018). In the daytime, $H_2SO_4$ and $H_2SO_4-NH_3$ clusters appear to be the most prominent negative ions (Schobesberger et al., 2015; Schobesberger et al., 2013). However, they have not yet been thoroughly studied regarding their appearance and their plausible links to atmospheric IIN.

Along with the changes in temperature and in ion concentration and composition, the importance of IIN is expected to vary considerably. In this study, we revisit the ion measurement in Hyytiälä, aiming to connect our current understanding of the formation of ion clusters to the significance of IIN, with a special focus on the fate of $H_2SO_4-NH_3$ clusters. We also extend our analysis to ions other than $H_2SO_4$ clusters, i.e., HOMs, and identify their role in IIN, in addition to other measured parameters on site. Finally, this study confirms the consistency between chamber findings and atmospheric observations, even though it seems that at least two separate mechanisms are alternatively controlling the IIN in Hyytiälä.

## 2    Materials and Methods

For this study, we used data collected at the Station for Measuring Forest Ecosystem-Atmospheric Relations (SMEAR II station), in Hyytiälä, Southern Finland (Hari and Kulmala, 2005). In this study, our data sets were obtained from intensive campaigns in 3 consecutive springs, 2011 – 2013. The exact time periods of the APi-TOF measurements are 22[nd] of March until 24[th] of May 2011, 31[st] March until 28[th] of April 2012, and 7[th] April until 8[th] of June 2013. For 134 days we were able to extend our analysis to include: i) ion composition and chemical characterization using the APi-TOF (Junninen et al., 2010), ii) particle and ion number size distribution using NAIS (e.g., Mirme and Mirme 2013), iii) concentrations of $H_2SO_4$ and HOMs measured by the chemical ionization atmospheric-pressure-interface time-of-flight mass spectrometer (CI-APi-TOF see, e.g., (Jokinen et al., 2012; Ehn et al., 2014; Yan et al.,

2016), and iv) other relevant parameters, e.g., $NH_3$ (Makkonen et al., 2014), temperature and
cloudiness (Dada et al., 2017).

## 2.1    Measurement of atmospheric ions

The composition of atmospheric anions was measured using the atmospheric-pressure-
interface time-of-flight mass spectrometer (APi-TOF) (Junninen et al., 2010). The instrument
was situated inside a container in the forest, direct sampling the air outside. To minimize the
sampling losses, we firstly drew the air at a larger flow rate within a wide tube (40 mm inner
diameter), and another 30-cm-long coaxial tube (10 mm outer diameter and 8 mm inner
diameter) inside the wider one was used to draw 5 L/min towards the APi-TOF, 0.8 L/min out
of which will enter through the pinhole. After entering the pinhole, the ions are focused and
guided through two quadrupoles and one ion lens, and finally and detected by the time-of-flight
mass spectrometer.
Different from the commonly used chemical-ionization mass spectrometer (CIMS), the APi-
TOF does not do any ionization, so it only measures the naturally charged ions in the sample.
In the atmosphere, the ion composition is affected by the proton affinity of the species:
Molecules with the lowest proton affinity are more likely to lose the proton and thus become
negatively charged after colliding many times with other species; similarly, molecules with the
highest proton affinity would probably become positively charged ions. In addition to the
proton affinity, the neutral concentration also plays a role in determining the ion composition
by affecting the collision frequency. Due to the limited ionization rate in the atmosphere, there
is always a competition between different species in taking the charges. For example, the
$H_2SO_4$ often dominates the spectrum in the daytime when it is abundant, while in the night-
time nitrate ions and its cluster with HOMs are always the prominent due to the rare chance to
collide with the $H_2SO_4$. Since the signal strength of an ion in the APi-TOF depends not only
on the abundance of the respective neutral molecules, but also on the availability of other
charge-competing species, it is very important to note that the APi-TOF can not quantify the
neutral species.
One important virtue of APi-TOF is that it does not introduce extra energy during sampling,
which ensures the sample is least affected when compared to other measurement techniques
such as CIMS. although fragmentation cannot be fully avoided inside the instrument
(Schobesberger et al., 2013). Because of this, it is a well-suited instrument to directly measure
the composition of weakly bonded clusters in the atmosphere.
The APi-TOF data were processed with the tofTools package (version 6.08) (Junninen et al.,
2010). Since the ion signal in APi-TOF is usually weak, a 5-hour integration time was used,
after which the signals of $H_2SO_4$-$NH_3$ clusters and HOMs were fitted (See Fig.1). For HOM
signals, we used the same peaks reported in Bianchi et al., (2017), and the total signal of HOM
ions is the sum of all identified HOMs.
It should also be mentioned that the voltage tuning of the instrument was not the same in the
years we analyzed, which led to differences in the ion transmission efficiency function. For
example, we noticed that in 2011, the largest $H_2SO_4$-$NH_3$ clusters contained 6 $H_2SO_4$ molecules,
whereas more than 10 $H_2SO_4$ were observed in the clusters in other years. This was very likely
due to the very low ion transmission in the mass range larger than about 700 Th for the
measurements in 2011. However, this should not affect our results and conclusions, because
clusters consisting of 6 $H_2SO_4$ molecules had little difference from larger clusters in affecting
the IIN in terms of occurrence probability (see more details in Sect. 3.3.1).
2.2   Measurement of $H_2SO_4$ and HOMs
The concentrations of $H_2SO_4$ and HOMs were measured by the chemical ionization
atmospheric-pressure-interface time-of-flight mass spectrometer (CI-APi-TOF). The details of
the quantification method for $H_2SO_4$ can be found in Jokinen et al., (2012) and for HOMs in
Kirkby et al., 2016. For all data, we applied the same calibration coefficient ($1.89 \times 10^{10}$ 1/$cm^3$)
reported by Jokinen et al., (2012).
Although the tuning of the CI-APi-TOF was not exactly the same during the measurement
period included in this study, no systematic difference was found in the concentrations of
$H_2SO_4$ and HOMs from different years.
2.3   Measurements of ion and particle size distribution
The mobility distribution of charged particles and air ions in the range 3.2-0.0013 $cm^2V^{-1}s^{-1}$
(corresponding to mobility diameter 0.8 – 42 nm) were measured together with the size
distribution of total particles in the range ~2.5 - 42 nm using a neutral cluster and air ion
spectrometer (NAIS, Airel Ltd., (Mirme and Mirme, 2013)). The instrument has two identical
differential mobility analyzers (DMA) which allow for the simultaneous monitoring of positive
and negative ions. In order to minimize the diffusion losses in the sampling lines, each analyzer
has a sample flow rate of 30 L $min^{-1}$ and a sheath flow rate of 60 L $min^{-1}$. In "particle mode",
when measuring total particle concentration, neutral particles are charged by ions produced
from a corona discharge in a "pre-charging" unit before they are detected in the DMAs. The
charging ions used in this process were previously reported to influence the total particle
concentrations below ~2 nm (Asmi et al., 2008; Manninen et al., 2010); for that reason, only
the particle concentrations above 2.5 nm were used in the present work. Also, each
measurement cycle, i.e. 2 min in ion mode and 2 min in particle mode, is followed by an offset
measurement, during which the background signal of the instrument is determined and then
subtracted from measured ion and particle concentrations. In addition, particle size
distributions between 3 and 990 nm were measured with a differential mobility particle sizer
(DMPS) described in details in Aalto et al., (2001). Based on earlier work by Kulmala et al.,
(2001), this data were used to calculate the condensation sink (CS), which represents the rate
of loss of condensing vapors on pre-existing particles.
2.4    Measurement of Meteorological parameter
The meteorological variables used as supporting data in the present work were measured on a
mast, all with a time resolution of 1 min. Temperature and relative humidity were measured at
16.8 m using a PT-100 sensor and relative humidity sensors (Rotronic Hygromet MP102H with
Hygroclip HC2-S3, Rotronic AG, Bassersdorf, Switzerland), respectively. Global radiation
was measured at 18 m with a pyranometer (Middleton Solar SK08, Middleton Solar, Yarraville,
Australia), and further used to calculate the cloudiness parameter, as done previously by Dada
et al., (2017, and references therein). This parameter is defined as the ratio of measured global
radiation to theoretical global irradiance, so that parameter values < 0.3 correspond to a
complete cloud coverage, while values > 0.7 are representative of clear sky conditions.

2.5    Calculation of particle formation rates and growth rates

The formation rate of 2.5 nm particles includes both neutral and charged particles, and it was
calculated from the following equation:
$$J_{2.5} = \frac{dN_{2.5-3.5}}{dt} + CoagS_{2.5} \times N_{2.5-3.5} + \frac{1}{1nm} GR_{1.5-3} \times N_{2.5-3.5} \qquad \text{Eq. 1}$$
where $N_{2.5-3.5}$ is the particle concentration between 2.5 and 3.5 nm measured with the NAIS in
particle mode, $CoagS_{2.5}$ is the coagulation sink of 2.5 nm particles derive$N^{\pm}_{2.5-3.5}$d from DMPS
measurements and $GR_{1.5-3}$ is the particle growth rate calculated from NAIS measurements in
ion mode. Calculating the formation rate of 2.5 nm ions, or charged particles includes two
additional terms to account for the loss of 2.5 − 3.5 nm ions due to their recombination with
sub-3.5 nm ions of the opposite polarity (fourth term of Eq. 2) and the gain of ions caused by
the attachment of sub-2.5 nm ions on 2.5-3.5 nm neutral clusters (fifth term of Eq. 2):
$$J_{2.5}^{\pm} = \frac{dN_{2.5-3.5}^{\pm}}{dt} + CoagS_{2.5} \times N_{2.5-3.5}^{\pm} + \frac{1}{1nm} GR_{1.5-3} \times N_{2.5-3.5}^{\pm} + \alpha \times N_{2.5-3.5}^{\pm} N_{<3.5}^{\mp} - \beta \times N_{2.5-3.5} N_{<2.5}^{\pm} \qquad \text{Eq.2}$$
where $N_{2.5-3.5}^{\pm}$ is the concentration of positive or negative ions between 2.5 and 3.5 nm, $N_{<2.5}^{\pm}$ is
the concentration of sub-2.5 nm ions of the same polarity and $N_{<3.5}^{\mp}$ is the concentration of sub-
3.5 nm ions of the opposite polarity, all measured with the NAIS in ion mode. $\alpha$ and $\beta$ are
the ion-ion recombination and the ion-neutral attachment coefficients, respectively, and were
assumed to be equal to $1.6 \times 10^{-6}$ cm$^3$ s$^{-1}$ and $0.01 \times 10^{-6}$ cm$^3$ s$^{-1}$, respectively. We consider
these values as reasonable approximations, keeping in mind that the exact values of both $\alpha$ and
$\beta$ depend on a number of variables, including the ambient temperature, pressure and relative
humidity as well as the sizes of the colliding objects (ion-ion or ion-aerosol particle) (e.g.
Hoppel, 1985; Tammet and Kulmala, 2005; Franchin et al., 2015).
$GR_{1.5-3}$ were calculated from NAIS data in ion mode using the "maximum" method introduced
by (Hirsikko et al., 2005). Briefly, the peaking time of the ion concentration in each size bin of
the selected diameter range was first determined by fitting a Gaussian to the concentration. The
growth rate was then determined by a linear least square fit through the times. The uncertainty
in the peak time determination was reported as the Gaussian's mean 67% confidence interval,
and was further taken into account in the growth rate determination.
A similar approach was used to estimate the early growth rate of the $H_2SO_4$-$NH_3$ clusters
detected with the APi-TOF. Prior to growth rate calculation, we first converted cluster masses
into diameters in order to get growth rate values in nm h$^{-1}$ instead of amu h$^{-1}$. For that purpose,
we applied the conversion from Ehn et al., (2011), using a cluster density of 1840 kg m$^{-3}$. The
time series of the cluster signals were then analysed in the same way as ion or particle
concentrations using the "maximum" method from Hirsikko et al. (2005), and the growth rate
was calculated using the procedure recalled above. Our ability to determine the early cluster
growth rate from APi-TOF measurement was strongly dependent on the strength of the signal
of the different $H_2SO_4$-$NH_3$ clusters. As a consequence, the reported growth rates characterize
a size range which might slightly vary between the events, falling in a range between 1 and 1.7
nm.

**3    Results and Discussion**
3.1    **Daytime ion composition**
We examined the daytime ion composition of 134 days from three consecutive springs (2011-
2013) in Hyytiälä. Consistent with the findings by previous studies, showing that $H_2SO_4$

clusters are the most abundant ions in the daytime (Ehn et al., 2010; Bianchi et al., 2017), we found that $NH_3$-free $H_2SO_4$ clusters can contain up to three $H_2SO_4$ molecules when counting the $HSO_4^-$ also as one $H_2SO_4$ molecule (($H_2SO_4)_2HSO_4^-$), and that $NH_3$ is always present in clusters containing 4 or more $H_2SO_4$ molecules. The latter feature suggests the important role of $NH_3$ as a stabilizer in growing $H_2SO_4$ clusters (Kirkby et al. 2011). $NH_3$-free clusters (at least dimers $H_2SO_4HSO_4^-$) were observed on 116 measurement days, but the signal intensity varied from day to day. Bigger clusters that contained $NH_3$ were observed on 39 days, containing a maximum of 4 to 13 $H_2SO_4$ per cluster. Figure 1 provides four examples of daytime ion spectra, including an $NH_3$-free case (Fig. 1A) and three cases with a different maximum size of $H_2SO_4$-$NH_3$ clusters (Fig. 1B-D), illustrating the significant variations in signal and maximum size of $H_2SO_4$-$NH_3$ clusters. In the $NH_3$-free case, a larger number of HOM clusters (green circles) was observed, indicating a competition between $H_2SO_4$ and HOMs in taking the charges. The largest detected cluster during the measurement was $(H_2SO_4)_{12}(NH_3)_{13}HSO_4^-$, which corresponds to a mobility-equivalent diameter of about 1.7 nm according to the conversion method (Ehn et al., 2011) and is big enough to be detected by particle counters. Since the observed formation of such large $H_2SO_4$-$NH_3$ clusters is essentially the initial step of IIN, we anticipate that the variation of $H_2SO_4$-$NH_3$ clusters will influence the occurrence of IIN.

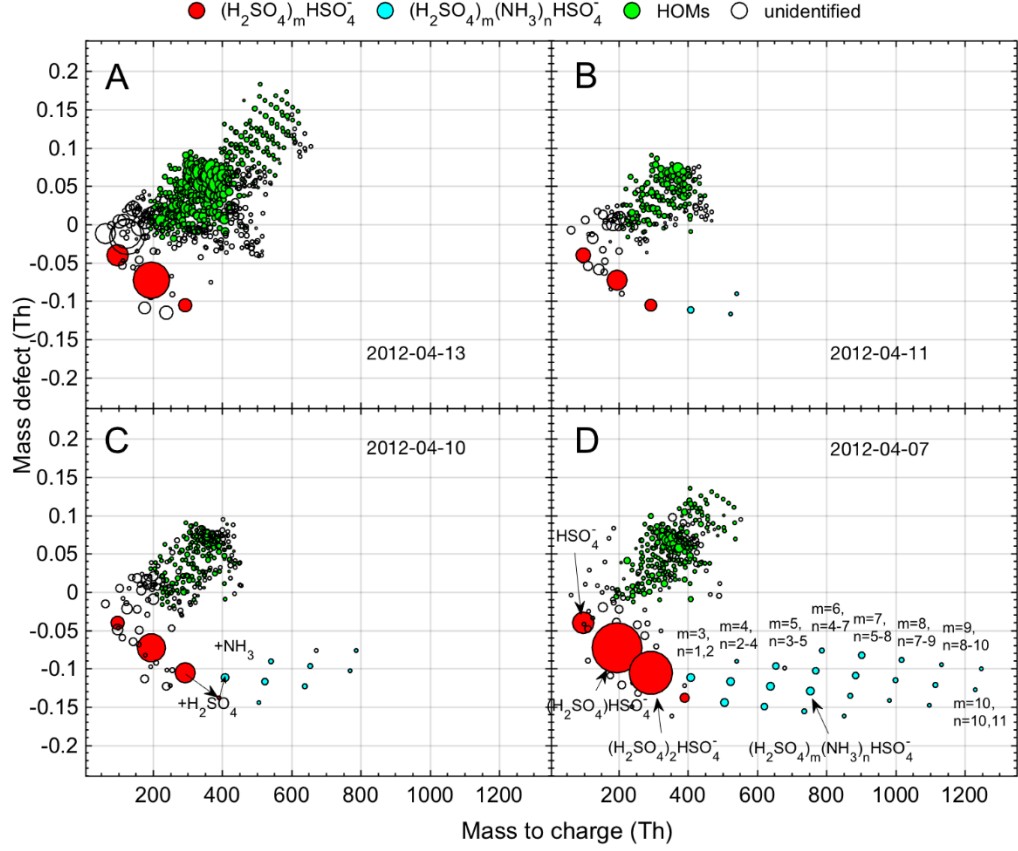

*Figure 1 Mass defect plot showing the composition of ion clusters on four separate days. A) NH₃-free clusters, B,C,D) H₂SO₄-NH₃ clusters with different maximum number of H₂SO₄ molecules. The circle size is linearly proportional to the logarithm of the signal intensity.*

## 3.2 **The determining parameters for $H_2SO_4$-$NH_3$ cluster formation**

To find out the dominating parameters that affect the formation of $H_2SO_4$-$NH_3$ clusters, we performed a correlation analysis that included the ambient temperature, relative humidity (RH), wind speed, wind direction, condensation sink (CS), as well as the gas-phase concentrations of $NH_3$, $H_2SO_4$, and HOMs. Among all the examined parameters, we found that the ratio between concentrations of HOMs and $H_2SO_4$ had the most pronounced influence on the appearance of $H_2SO_4$-$NH_3$ clusters. As shown in Figure 2, all $H_2SO_4$-$NH_3$ clusters were detected when [HOMs]/[$H_2SO_4$] was smaller than 30. No such dependence was observed for only [HOMs] or [$H_2SO_4$]. This implies that the appearance of $H_2SO_4$-$NH_3$ clusters is primarily controlled by the competition between $H_2SO_4$ and HOMs in getting the charges. More specifically, $HSO_4^-$, the main charge carrier in the daytime, may either collide with neutral $H_2SO_4$ to form large clusters to accommodate $NH_3$, or collide with HOMs that prevents the former process. In addition, a reasonable correlation was found between [HOMs]/[$H_2SO_4$] and temperature, likely

explained by emission of volatile organic compounds (VOC) increasing with temperature,
leading to higher HOMs concentrations, whereas the formation of $H_2SO_4$ is not strongly
temperature-dependent. This observation indicates that the formation of $H_2SO_4$-$NH_3$ clusters
may vary seasonally: we expect to see them more often in cold seasons when HOM
concentrations are low, and less often in warm seasons.
Parameters other than [HOMs]/[$H_2SO_4$] and temperature seemed to have little influence on the
formation of $H_2SO_4$-$NH_3$ clusters. Interestingly, we found that $NH_3$ was even lower when
$H_2SO_4$-$NH_3$ clusters were observed, indicating that the $NH_3$ concentration is not the limiting
factor for forming $H_2SO_4$-$NH_3$ clusters (also see section 3.4). In addition, $H_2SO_4$-$NH_3$ clusters
were observed in a wide range of RH spanning from 20 to 90 %, suggesting that RH is not
affecting the cluster formation. Besides, no clear influence from condensation sink (CS), wind
speed, or wind direction was observed.

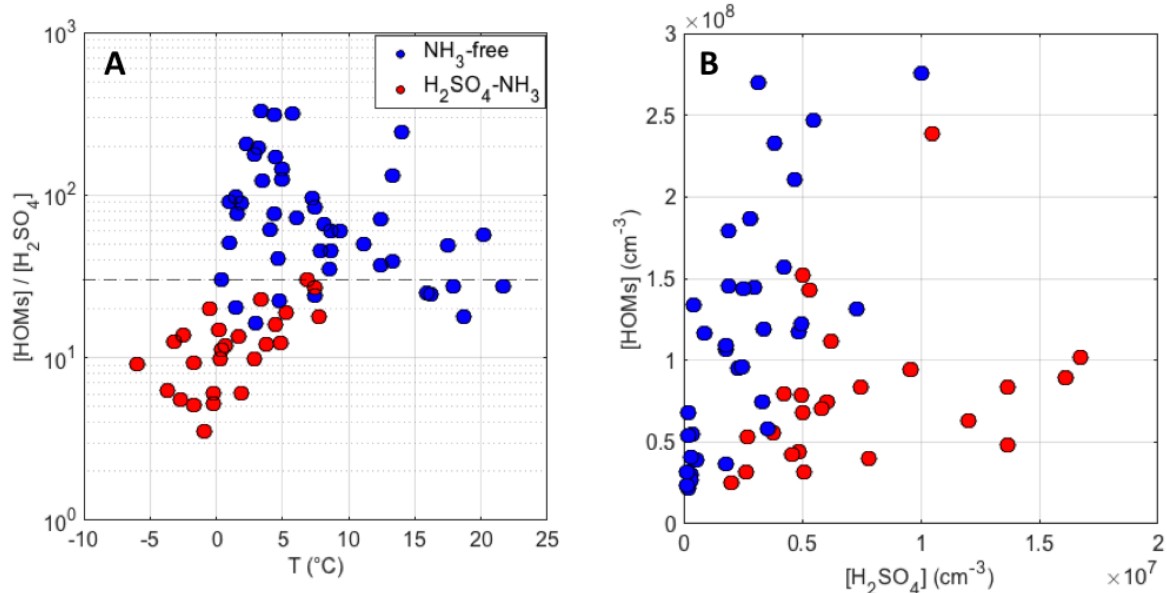


*Figure 2 The effect of concentration of HOMs, $H_2SO_4$, their ratio ([HOM]/[$H_2SO_4$]), and temperature*
*on the appearance of $H_2SO_4$-$NH_3$ clusters.*

### 3.3   The relation between $H_2SO_4$-$NH_3$ clusters and IIN

#### 3.3.1   The effect of cluster size on the probability of IIN events

We identified IIN events using data from the NAIS (ion mode) by observing an increase in the
concentration of sub-2 nm ions (Rose et al., 2018), and classified 67 IIN events out of the 134
days of measurements. We defined the IIN probability as the number of days when IIN events
were identified out of the total number of days that were counted. For example, the overall IIN
probably is 50 % (67 out of 134 days). We found that the maximum observed size of $H_2SO_4$-
$NH_3$ clusters may affect the occurrence of IIN. Our conclusion is complementary to previous
theories which stated that the critical step of particle nucleation is the formation of initial
clusters that are big enough for condensational growth to outcompete evaporation (Kulmala et
al., 2013). To further understand the size-dependency of IIN probability, we investigated the
IIN probability when different maximum sizes of $H_2SO_4$-$NH_3$ clusters were observed. As
illustrated in Figure 3, the IIN probability increases dramatically when larger $H_2SO_4$-$NH_3$
clusters were observed: IIN events were never observed when only $HSO_4^-$ or $H_2SO_4HSO_4^-$ were
present, whereas the IIN probability increased to about 50 – 60 % when the largest clusters
contained 3 – 5 $H_2SO_4$ molecules. IIN occurred in 24 out of 25 days (96 %) when the largest
clusters consisted of no less than 6 $H_2SO_4$ molecules. Thus, it is evident that the occurrence of
IIN is related to the size and thus the stability of $H_2SO_4$-$NH_3$ clusters, and that a cluster
consisting of 6 $H_2SO_4$ molecules seems to lie on the threshold size of triggering nucleation.

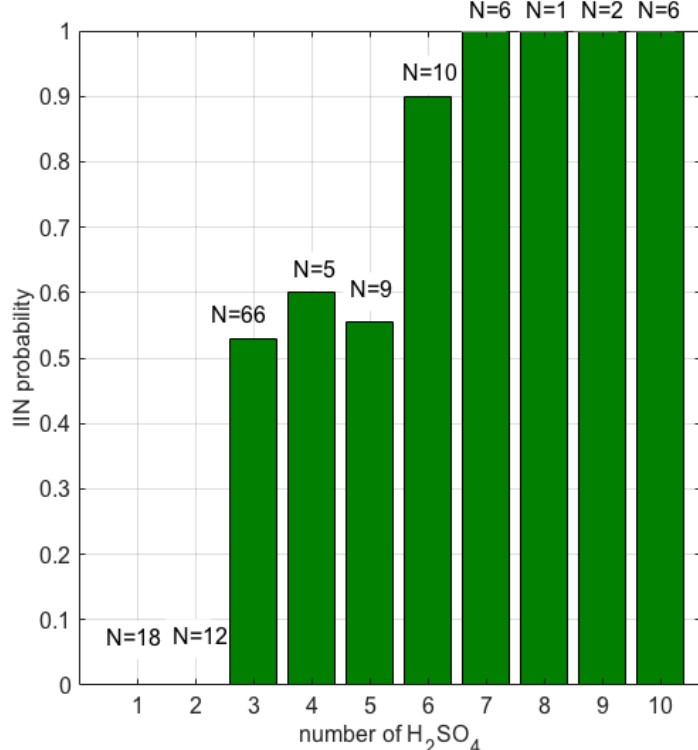


*Figure 3 The maximum number of $H_2SO_4$ molecules observed in clusters and the respective IIN*
*probability. The days when it was unclear if IIN occurred was counted as non-event days. N denotes*
*the number of days when such clusters were the largest observed.*
3.3.2    Continuous growth from clusters to 3 nm particles
Although the strong connection between the size of $H_2SO_4$-$NH_3$ clusters and the occurrence of
IIN was confirmed, it is challenging to directly observe the growth of these clusters in the
atmosphere, limited by the inhomogeneity of the ambient air and low concentrations of
atmospheric ions. Combining APi-TOF and NAIS measurements, we were able to follow the
very first steps of the cluster growth for 8 of the detected events. In Figures 4A and 4B, we
present two examples in which the continuous growth of $H_2SO_4$-$NH_3$ clusters to 3 nm (mobility
diameter) particles was directly evaluated using the maximum-time method. The maximum
times, determined from APi-TOF and NAIS data independently, fall nicely on the same linear
fit. The continuity of the growth and the linearity of the fit suggests that the current mechanism
($H_2SO_4$-$NH_3$, acid-base) explains the formation and growth of sub-3 nm ion clusters in these
cases. In most cases, the calculation of cluster GR from APi-TOF measurement suffered from
high uncertainties, but a weak positive correlation can be observed between the cluster growth
rate and $H_2SO_4$ concentration (Fig. 4C). This correlation is likely due to the collision of $H_2SO_4$
with existing $H_2SO_4$-$NH_3$ clusters being the limiting step for cluster growth when $NH_3$ is
abundant enough to follow up immediately (Schobesberger et al., 2015).

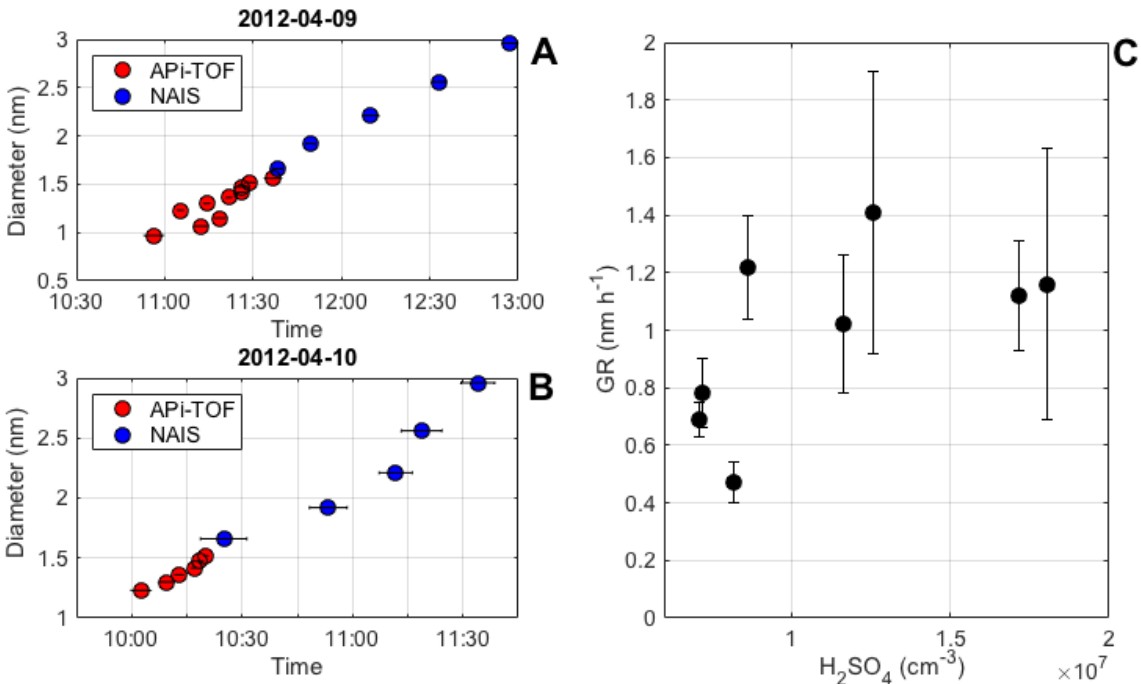


*Figure 4 Cluster growth rate determined from APi-TOF (A) and NAIS (B) measurements using the*
*maximum time method, and the correlation between growth rates and concentrations of $H_2SO_4$*
*molecules (C).*
3.4    Evidence for other IIN mechanisms
For the 134 days of measurements, we were able to identify 67 IIN events using the NAIS data,
out of which $H_2SO_4$-$NH_3$ clusters were observed on 32 days, implying that at least 35 IIN
events were likely driven by mechanism(s) other than $H_2SO_4$-$NH_3$. In Figure 5, we classified
the days according to the types of IIN observation: 32 IIN events involving $H_2SO_4$-$NH_3$ (S-E),
3 non-events with the presence of $H_2SO_4$-$NH_3$ clusters (S-NE), 35 IIN events involving other
mechanisms (O-E), 41 other non-event days (O-NE), and 23 days with unclear types. We
further present the respective statistics of additional measurements for the first four types of
days, including the concentrations of plausible precursor vapors, condensation sink and
meteorological parameters. It should be noted that the SA-NE has only 3 days, thus the statistics
on this type of days might not be fully representative.
Consistent with the previous discussion (Fig. 2), low temperatures are conducive of IIN events
via the $H_2SO_4$-$NH_3$ mechanism whilst being the highest other type of events (O-E) (Fig. 5A).
The clear-sky parameter (100% = clear sky and 0% = cloudiness) shows a noticeably higher
value during both event types compared to the non-event cases (Fig. 5B), indicating that photo-
chemistry related processes are important for all events. Moreover, the CS is obviously lower
for both types of events than on non-event days (Fig. 5C). Although a strong effect of CS on
the appearance of $H_2SO_4$-$NH_3$ clusters has not been noticed, it is a most important parameter
in regulating the occurrence of IIN. Similar effects of cloudiness and CS on governing the
occurrence of NPF have been reported by Dada et al., (2017) based on long-term data sets.
Remarkably, $NH_3$ has very low concentrations during $H_2SO_4$-$NH_3$ events in comparison to the
other type of events (Fig. 5D). This is likely due to high $NH_3$ concentrations coinciding with
higher temperature and thus elevated HOMs concentration, or the lower stability of $H_2SO_4$-
$NH_3$ clusters at high temperatures that can evaporate $NH_3$ back to the atmosphere. This
observation rules out the addition of $NH_3$ as a limiting step in the $H_2SO_4$-$NH_3$ nucleation
mechanism, but the participation of $NH_3$ in the other type of events cannot be excluded.
$H_2SO_4$ has the highest concentrations during the $H_2SO_4$-$NH_3$-involved events (Fig. 5E), but the
concentration of $H_2SO_4$ in S-NE days is not much lower, suggesting that the occurrence of
$H_2SO_4$-$NH_3$-involved events is not solely controlled by the $H_2SO_4$ concentration. The
Incorporating the effect of CS ($[H_2SO_4]$/CS) significantly improves the separation (Fig.5F).
McMurry and coworkers (Mcmurry et al., 2005) have introduced a parameter L (Eq.3) to
quantitatively evaluate the likelihood of NPF, and they found that NPF mostly occurred when
L is smaller than 1. A similar result has been reported by Kuang et al., (2010), and a slightly
different threshold L value of 0.7 was determined.
$$L = \frac{CS}{[H_2SO_4]} \cdot \frac{1}{\beta_{11}} \text{ (Eq.3)}$$
Here, L is dimensionless parameter representing the probability that NPF will not occur, and
$\beta_{11}$ is the collision rate between $H_2SO_4$ vapor molecules, which is characterized as $4.4\times 10^{-10}$
$cm^3s^{-1}$. Our results suggest a consistent L that most (75 percentile) S-E cases happen when L
is lower than 0.73 and most (75 percentile) S-NE cases are observed when L is larger than 1.54.
HOM concentrations are highest in the case of other events, revealing that HOMs play a key
role in this mechanism (Fig. 5F), although the contribution of $H_2SO_4$ in this HOM-involving
IIN mechanism cannot be excluded. Similar to the $H_2SO_4$-$NH_3$-driven cases, incorporating the
CS better distinguishes the event and non-event cases.
Overall, our results suggest that the concentrations of $H_2SO_4$ and HOMs, together with the CS
governs the occurrence of IIN, whereas their ratio determines the exact underlying mechanism
(Fig. 2). Although $H_2SO_4$-$NH_3$ and HOMs clearly drives the S-E and O-E events, respectively,
we cannot exclude the later participation of HOMs in SA-E cases or $H_2SO_4$ in O-E cases.
Different NPF mechanisms have also been identified at the Jungfraujoch station (Bianchi et
al., 2016) Frege et al., 2018) when influenced by different air masses. At SMEAR II station,
on the other hand, our results suggest that the natural variation of temperature is already
sufficient to modify the NPF mechanism via modulating the biogenic VOC emissions.

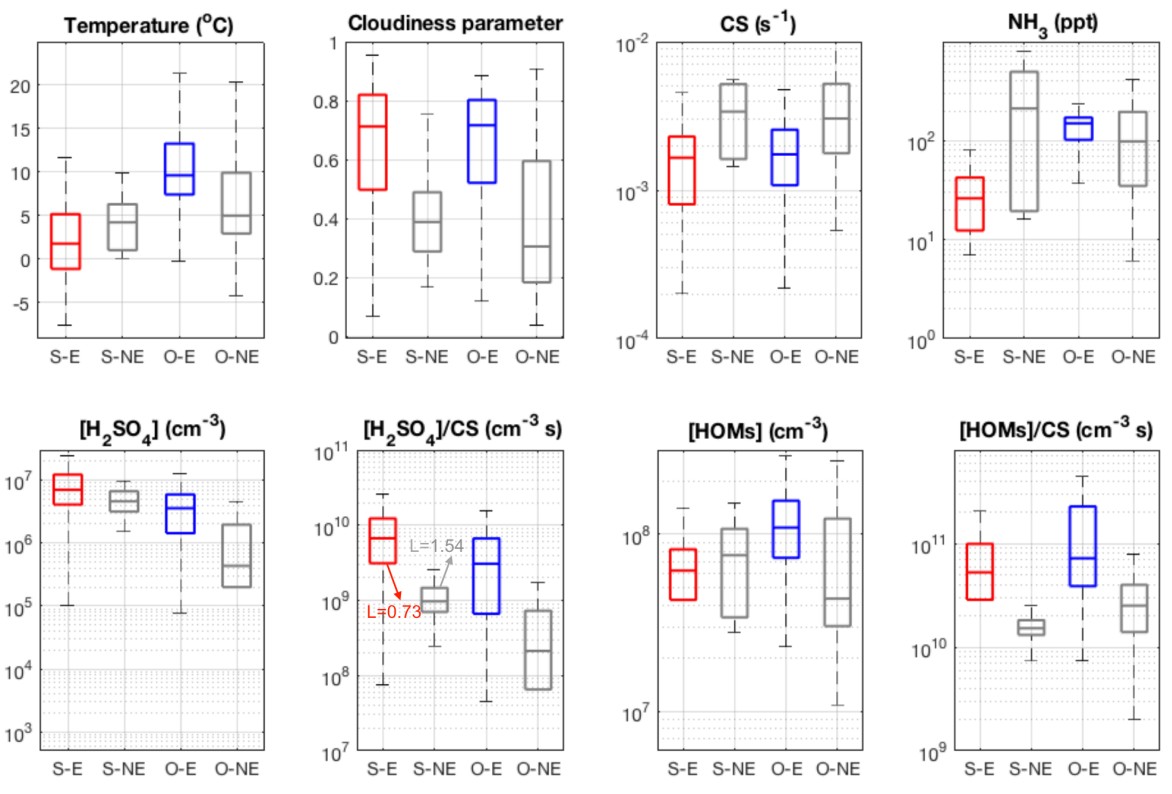


Figure 5 Comparison of different parameters for $H_2SO_4$-$NH_3$-involved events (S-E, red bars), non-events with the presence of $H_2SO_4$-$NH_3$ clusters (S-NE, first column of black bars), other events (O-E, blue bars), and other non-events (O-NE, second column of black bars).

3.5  Contribution of IIN to total nucleation rate

In order to get further insight into the importance of IIN during our measurements, we compared the formation rate of 2.5 nm ions, $J_{ION} = J_{2.5}^{\pm}$ (see Eq.2), to the total formation rate of 2.5 nm particles, $J_{TOT} = J_{2.5}$ (see Eq.1). The ratio $J_{ION}/J_{TOT}$ is equal to the charged fraction of the 2.5 nm particle formation rate. In analyzing field measurements, a similar ratio at a certain particle size (typically 2 nm) has commonly been used to estimate the contribution of ion-induced nucleation to the total nucleation rate (see Hirsikko et al. 2011 and references therein). It should be noted that $J_{ION}/J_{TOT}$ represents only a lower limit for the contribution of ion-induced nucleation, as this ratio does not take into account the potential neutralization of growing charged sub-2.5 nm particles by ion-ion recombination (e.g. Kontkanen et al., 2013; Wagner et al., 2017). At present, measuring the true contribution of ion-induced nucleation to the total nucleation rate is possible only in the CLOUD chamber (Wagner et al., 2017).

We were able to calculate $J_{ION}$ and $J_{TOT}$ for 57 (out of 67) cases, and the ratio $J_{ION}/J_{TOT}$ varied from 4 to 45%, showing a clear correlation with the HOM signal (Fig. 6A). This indicates the participation of HOMs even in $H_2SO_4$-$NH_3$-driven cases. In addition, most of the high $J_{ION}/J_{TOT}$ ratios were observed at moderate or low $H_2SO_4$ concentrations, e.g., $J_{ION}/J_{TOT} > 15$ % was only observed when $[H_2SO_4] < 6 \times 10^6$ cm$^{-3}$. These observations indicate that HOMs are important in high $J_{ION}/J_{TOT}$ cases, while during events driven by $H_2SO_4$-$NH_3$ clusters low $J_{ION}/J_{TOT}$ is more often observed. Accordingly, the median value of $J_{ION}/J_{TOT}$ for the $H_2SO_4$-$NH_3$ cases is about 12 % and is clearly higher (18 %) in HOM-driven events (Fig. 6D). Figures 6B and 6C reveal that both $J_{ION}$ and $J_{TOT}$ values are in fact higher in $H_2SO_4$-$NH_3$ cases, but the neutral nucleation pathway is relatively more enhanced, leading to the lower ratio. These results suggest that ion-induced nucleation plays a more important role in the events driven by HOMs than in the events driven by $H_2SO_4$-$NH_3$. A plausible explanation is that $NH_3$ is performing well in stabilizing $H_2SO_4$ molecules during the clustering process, whereas ions are a relatively more important stabilizing agent for HOM clustering.

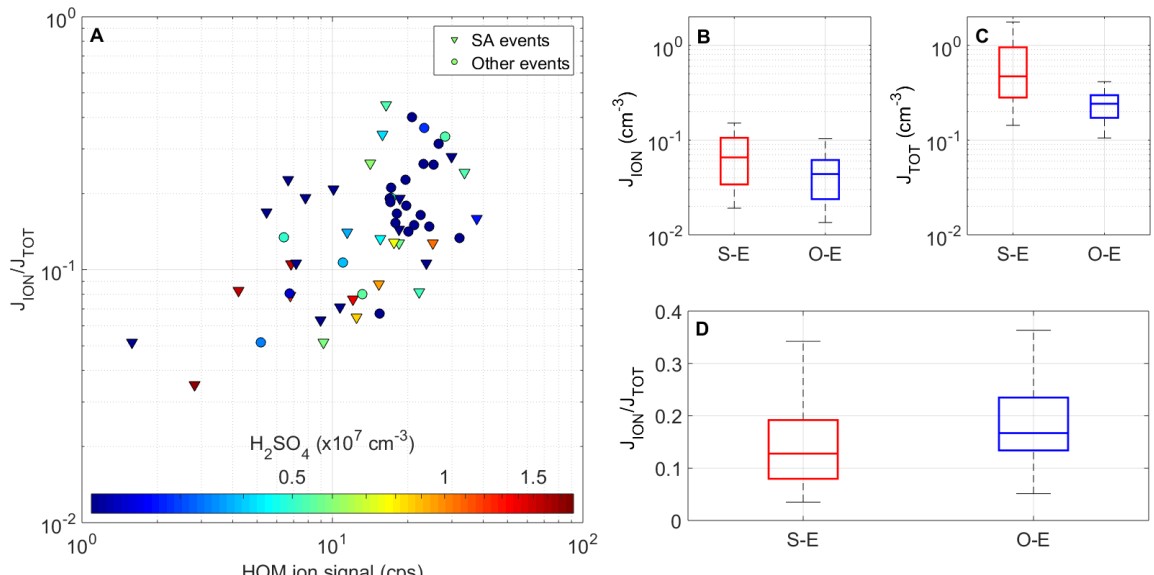

409

*Figure 6. Formation rate 2.5 nm ions and total particles (both ions and neutral clusters) under different nucleation mechanisms. A) Charged fraction of the formation rate of 2.5 nm particles as a function of the total signal of HOM ions color-coded by the $H_2SO_4$ concentration, and (B, C and D) the differences in $J_{ION}$, $J_{TOT}$, and $J_{ION}/J_{TOT}$ between the $H_2SO_4$–$NH_3$-involved events (S-E) and other events (O-E).*

## 4   Summary

We investigated the formation of $H_2SO_4$-$NH_3$ anion clusters measured by APi-TOF during three springs from 2011 to 2013 in a boreal forest in Southern Finland and their connection to IIN. The abundance and maximum size of $H_2SO_4$-$NH_3$ clusters showed great variability. Out of the total 134 measurement days, $H_2SO_4$-$NH_3$ clusters were only seen during 39 days. The appearance of these clusters was mainly regulated by the concentration ratio between HOMs and $H_2SO_4$, which can be changed by temperature via modulating the HOM production.

We found that the maximum observable size of $H_2SO_4$-$NH_3$ clusters has a strong influence on the probability of an IIN event to occur. More specifically, when clusters containing 6 or more $H_2SO_4$ molecules were detected, IIN was observed at almost 100% probability. We further compared the cluster ion growth rates from APi-TOF and NAIS using the maximum-time method. In these $H_2SO_4$-$NH_3$ driven cases when we could robustly define the track of the cluster evolution, the cluster growth was continuous and near linear for cluster-sizes up to 3 nm, suggesting co-condensation of $H_2SO_4$ and $NH_3$ as the sole growth mechanism. This does not exclude that organics could also participate in the growth process in Hyytiälä on other days. In addition, we noticed that there was a mechanism driving the IIN, and HOMs are the most likely responsible species, although $H_2SO_4$ and $NH_3$ might also participate in this mechanism.

Such mechanism was responsible for at least 35 IIN events during the measurement days and
is expected to be the prevailing one in higher-temperature seasons.
The contribution of IIN to the total rates of NPF differs between events driven by $H_2SO_4$-$NH_3$
and by HOMs. IIN plays a bigger role in HOM-driven events, likely due to a relatively stronger
stabilizing effect of ions. Since the production of HOMs and $H_2SO_4$ are strongly modulated by
solar radiation and/or temperature, a seasonal variation of IIN can be expected, not only in
terms of frequency, but also in terms of the underlying mechanisms, and hence in terms of the
enhancing effect of ions. This information should be considered in aerosol formation modelling
in future works.

**Acknowledgement**
This work was partially funded by Academy of Finland (1251427, 1139656, 296628, 306853,
Finnish centre of excellence 1141135), the EC Seventh Framework Program and European
Union's Horizon 2020 program (Marie Curie ITN no. 316662 "CLOUD-TRAIN", no. 656994
"Nano-CAVa", no. 227463 "ATMNUCLE", no. 638703 "COALA", no 714621
"GASPARCON", and no.742206 "ATM-GTP"), European Regional Development Fund
project "MOBTT42". We thank the tofTools team for providing tools for mass spectrometry
analysis.

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
