# Peer review of "The role of H2SO4-NH3 anion clusters in ion-induced aerosol nucleation"

_Atmospheric Chemistry and Physics, 2018_

## Referee Comment (RC1) · Anonymous Referee #1 · 23 Apr 2018

Review of

The role of $H_2SO_4$-$NH_3$ anion clusters in ion-induced aerosol nucleation mechanisms in the boreal forest

by

Chao Yan, Lubna Dada, Clémence Rose, Tuija Jokinen, Wei Nie, Siegfried Schobesberger, Heikki Junninen, Katrianne Lehtipalo, Nina Sarnela, Ulla Makkonen, Olga Garmash, Yonghong Wang, Qiaozhi Zha, Pauli Paasonen, Federico Bianchi, Mikko Sipilä, Mikael Ehn, Tuukka Petäjä, Veli-Matti Kerminen, Douglas R. Worsnop, Markku Kulmala

This paper analyzes data from springtime observations at the Hyytiälä observatory carried out in 2011, 2012, and 2013. Measurements of ion chemical composition were made using the Atmospheric Pressure interface Time of Flight Mass Spectromter (APi-TOF), while measurements of precursor vapor concentrations ($H_2SO_4$, and HOM) were measured by chemical ionization mass spectrometry. $NH_3$ was also measured. In addition, a DMPS system was used to measure aerosol size distributions (3-990 nm), and neutral particle (2.5-42 nm) and ion (0.8-42 nm) number distributions were measured with the NAIS. These data were used to calculate particle growth rates and ion induced and neutral particle nucleation rates. Some evidence for ion induced nucleation (IIN) was found on about 50% of measurement days, although IIN rates were less than neutral particle nucleation rates. When IIN was observed, anions comprised of $H_2SO_4$-$NH_3$ were dominant on about 50% of the time. However, when [HOM]/[$H_2SO_4$] exceeded 30, the dominant IIN pathway appeared to involve HOM. This is a nice study that provides valuable new insights into the chemical processes that lead to IIN at this location. It is significant work based on excellent observational results. The paper is concise and clearly written. I recommend publication in ACP, and offer a few suggestions for the authors to consider.

Suggestion:
On p. 11 the authors point out that the likelihood of $H_2SO_4$-$NH_3$ IIN depends strongly on [$H_2SO_4$]/CS. When this ratio equaled about $10^{10}$ cm$^{-3}$s, IIN was observed, while IIN was far less likely when [$H_2SO_4$]/CS=$10^9$ cm$^{-3}$s. McMurry and coworkers (McMurry et al. 2005) argued that the likelihood that nucleated clusters will grow into new particles decreases rapidly when the dimensionless parameter, $L$, exceeds a value on the order of 1. Kuang and coworkers (Kuang et al. 2010) showed that new particle formation was rarely observed when $L>0.7$. From equations [A3] and [A5] of McMurry et al (2005), it is straightforward to show that:

$$L = \frac{CS}{[H_2SO_4]} \cdot \frac{1}{\beta_{11}}$$

where $\beta_{11}$ is the collision rate between $H_2SO_4$ vapor molecules. A characteristic value for $\beta_{11}$ is $4.4 \times 10^{-10}$ cm$^3$s$^{-1}$. It follows than for these data, $H_2SO_4$-$NH_3$ IIN was observed for L~0.22 but not for L~2.2. This is consistent with theoretical expectations and prior work, and would move the authors closer to providing a quantitative theoretical explanation for the S IIN results shown in Figure 5F.

Points that should be clarified.

•Line 26, p. 1: "All such clusters were observed..." In the context of this sentence, this implies that all clusters from #S=3 to infinity were observed. This is obviously not what is intended.

• p. 4, line 120: "...a best instrument..." The previous sentence acknowledges that some fragmentation occurs. It might also be mentioned that the extent of fragmentation is not quantitatively understood. Have the authors confirmed that the APi-TOF produces less fragmentation that other mass spectrometers with different interface designs? I don't think so.

Minor editorial corrections:
The paper contains numerous minor, distracting language errors, which I illustrate with the following examples. The text should be thoroughly edited by a native English speaker.

•p. 2, line 57: should be "as ion-induced..."
•p. 4, lines 111-112: "..often dominates the daytime spectrum in the daytime when it is abumdant,..." ???
•p. 4, line 118: replace "comparing" with "compared"
•p. 4, line 127: delete ","
•p. 5, line 163: delete "In specific,"
•p. 11, Figure 3 caption: replace "unclear is IIN..." with "unclear if IIN..."
•p. 11, line 285: delete "however" (Alternatively, it could be separated using commas, but this would make for an awkward sentence.)
•p. 12, line 309: "this type of days..."
•p. 12, line 310: "are conduce of IIN events..."
•p. 12, line 315: "has not been evidenced,..."
•p. 14, Figure 5: Difficult to distinguish blue from black boxes.
•p. 14, line 349: "This indicate the..."
•p. 15, line 371: "The abundancy and ..."

Kuang, C., I. Riipinen, T. Yli-Juuti, M. Kulmala, A. V. McCormick and P. H. McMurry (2010). "An improved criterion for new particle formation in diverse atmospheric environments." *Atmospheric Chemistry and Physics* **10**: 1-12. doi: 10.5194/acp-10-1-2010.

McMurry, P. H., M. A. Fink, H. Sakurai, M. R. Stolzenburg, L. Mauldin, K. Moore, J. N. Smith, F. L. Eisele, S. Sjostedt, D. Tanner, L. G. Huey, J. B. Nowak, E. Edgerton and D. Voisin (2005). "A Criterion for New Particle Formation in  the Sulfur-Rich Atlanta Atmosphere." *Journal of Geophysical Research - Atmospheres* **110**: D22S02. doi: 10.1029/2005JD005901.

---

## Referee Comment (RC2) · Anonymous Referee #2 · 14 Jun 2018

**Review of Yan et al. acp-2018-187**

The manuscript analyzes cluster ion data from the SMEAR station in Hyytiala, Finland, from three springtime measurement periods. Data from anion measurements with an APi-TOF mass spectrometer and an NAIS instrument are analyzed. The focus of the analysis is on H2SO4-NH3 cluster ions in comparison to HOM ions and their relation to aerosol nucleation events. It is found that the ratio between [HOMs] and [H2SO4] controls the presence of large H2SO4-NH3 clusters. Furthermore, the probability for IIN to occur is largest and reaching almost 100% when clusters containing 6 or more H2SO4 molecules are present. The contribution of IIN to the total nucleation is reported to range between 4 and 45%, with an average of 12% contribution for cases that are dominated by H2SO4-NH3 nucleation and 18% in HOM-driven events.

The manuscript is an extension of a series of papers focusing on results from the APi-TOF measurements in Hyytiala (e.g. Ehn et al., 2010 and 2011, Schobesberger et al., 2013 and 2015; Yan et al., 2016; Bianchi et al., 2017). The previous papers focused mostly on the role of HOMs while this one focuses on the role of H2SO4-NH3 anion clusters, therefore the paper presents sufficient new material to warrant publication in ACP.

There is a number of minor points and technical corrections to consider before publication in ACP:

**Minor points**

line 43: The paper by Dunne et al., Science, 2016 should be referenced here as well.

136: quantification of the APi-TOF results. Was the transmission of the APi-TOF characterized as described by Heinritzi et al., AMT, 2016? Can you be sure that the transmission did not change due to the changes in tuning (l 139)?

181: What about recombination with ions larger than 3.5 nm?

Figure 2: panel B is as important as panel A. Why is B just shown as a small inset? Please show B as a separate panel of the same size as A, or even as a separate Figure.

347: Please explain in detail how $J_{IIN}$ for 2.5 nm particles was calculated (here, or in Section 2).

404-575: **Please check all references carefully**: In many cases there are co-authors missing (and no "et al." is included), e.g. Bianchi et al., 2017, Dada et al., 2017, Ehn et al., 2010, Ehn et al., 2011, Kulmala et al., 2004, Schobesberger et al., 2013 and 2015, and even in Yan et al., 2016, and many others.

General comment on choice of cited references: There is no doubt that the Kulmala group has produced lots of important research with respect to ground-based cluster ion composition measurements with the APi-TOF in Hyyttiälä, and it is therefore ok to reference the previous work of your own group frequently. Nevertheless, there have been various contributions to the field of H2SO4-NH3-IIN by other groups and the choice of references discussed for example in the introduction seems somewhat unbalanced. Out of the 34 references listed in the references section, 29 are from the Kulmala group or co-authored by the Kulmala group (and the 5 remaining references are mainly general ones such as reviews or the IPCC report). It is expected in scientific publications to give reference also to the previous work by others that is relevant for your work. Therefore I suggest to mention/discuss also work from other groups, e.g. Eisele et al., JGR, 2006; Iida et al., JGR 2006, Tammet et al., Atm. Res. 2014; Rose et al., ACP, 2013; Boulon et al., ACP 2010; Kurten et al., JGR, 2016; Froyd and Lovejoy, JPC, 2011, etc. to give some credit also to the rest of the scientific world

that performed measurements of H2SO4-NH3 ion induced nucleation and other ion clusters. Also  Bianchi et al., Science, 2016; Dunne et al., Science, 2016, and Wagner et al., ACP, 2018, should be included and discussed in the context of this paper (I recognize that these are also co-authored/authored by the Helsinki group).

**Technical corrections:**

line 65 and 67. Bianchi et al. is referenced twice, one time within a sentence is sufficient.

75: understandings → understanding

92: insert space between semicolon and Ehn, as well as between semicolon and Yan

111: "daytime spectrum in the daytime…" → avoid duplication

113: "of an ion in *the* APi-TOF…"

115: "note that *the* APi-TOF…"

119: instruments → instrument

120: "is a best instrument" → "is a good…" or "is a well-suited…."

127: not the same in *the three* years…

129: "clusters contained 6 clusters" → "clusters contained 6 SA molecules"

130: "in the clusters were observed" → were observed in the clusters

131: larger than 700 Th *for the measurements in 2011.*

133: Figure 3 is called here before Figure 2 is called. → Change order of Figures 2 and 3.

139: tunings of CI-APi-TOF → *tuning* of *the* CI-APi-TOF

181, 184: "Eq. S2" → change numbering of the Equation to "Eq. 2"

197: *A similar* approach

204: measurements  → measurement

204: dependant → dependent

242: prevents → suppresses

261: "by observing an increase…" (delete "of")

262: "sub-2nm ions", I think you mean "ions *larger* than 2 nm" here?

279:  "…unclear is IIN occurred was counted…" → "…unclear *if* IIN occurred *were* counted…"

289: permanence → continuity

319 and 324: *the* other type of events

321: less → lower

322: clusters at high temperatures that *can evaporate* NH3 back to the atmosphere.

340: Figure 5, panel B: "cloudiness parameter" should probably be "clear-sky parameter", or it should be explained that 1 = clear-sky = 0% cloudiness; and 0 = 100% cloudiness

349: This *indicates*

361: Figure 6, panels B and C could be depicted with identical y-range (e.g. 10^-2 to 20), then a comparison would be easier. At least some tick marks should be added to panels B and C.

368 "Summary and Conclusions"  → "Summary" (there are no new conclusions, it is just a summary of the findings presented previously)

380: from → for

382: *on* other days

385: *a* mechanism

385: at least responsible for → responsible for at least

404: Reference → References

---

## Author Comment (AC1) · 2 Jul 2018

This paper analyzes data from springtime observations at the Hyytiälä observatory carried out in 2011, 2012, and 2013. Measurements of ion chemical composition were made using the Atmospheric Pressure interface Time of Flight Mass Spectrometer (APi-TOF), while measurements of precursor vapor concentrations (H2SO4, and HOM) were measured by chemical ionization mass spectrometry. NH3 was also measured. In addition, a DMPS system was used to measure aerosol size distributions (3-990 nm), and neutral particle (2.5-42 nm) and ion (0.8-42 nm) number distributions were measured with the NAIS. These data were used to calculate particle growth rates and ion induced and neutral particle nucleation rates. Some evidence for ion induced nucleation (IIN) was found on about 50% of measurement days, although IIN rates were less than neutral particle nucleation rates. When IIN was observed, anions comprised of H2SO4-NH3 were dominant on about 50% of the time. However, when [HOM]/[H2SO4] exceeded 30, the dominant IIN pathway appeared to involve HOM. This is a nice study that provides valuable new insights into the chemical processes that lead to IIN at this location. It is significant work based on excellent observational results. The paper is concise and clearly written. I recommend publication in ACP, and offer a few suggestions for the authors to consider.

We would like to thank the referee for the suggestions and careful editorial comments. We reply to the comments item by item below (text in blue):

Suggestion:
On p. 11 the authors point out that the likelihood of H2SO4-NH3 IIN depends strongly on [H2SO4]/CS. When this ratio equaled about $10^{10}$ cm$^{-3}$s, IIN was observed, while IIN was far less likely when [H2SO4]/CS=109 cm-3s. McMurry and coworkers (McMurry et al. 2005) argued that the likelihood that nucleated clusters will grow into new particles decreases rapidly when the dimensionless parameter, L, exceeds a value on the order of 1. Kuang and coworkers (Kuang et al. 2010) showed that new particle formation was rarely observed when L>0.7. From equations [A3] and [A5] of McMurry et al (2005), it is straightforward to show that:

$$L = \frac{CS}{[H_2SO_4]} \cdot \frac{1}{\beta_{11}}$$

where β11 is the collision rate between H2SO4 vapor molecules. A characteristic value for β11 is 4.4e-10 cm$^3$s$^{-1}$. It follows than for these data, H2SO4-NH3 IIN was observed for L~0.22 but not for L~2.2. This is consistent with theoretical expectations and prior work, and would move the authors closer to providing a quantitative theoretical explanation for the S-E IIN results shown in Figure 5F.

$\beta_{11} = 4.4 \times 10^{-10}$ cm$^3$s$^{-1}$

Indeed, adding the discussion on the parameter L will move the observation towards more quantitative. We have added the following discussion:

"McMurry and coworkers (Mcmurry et al., 2005) have introduced a parameter L (Eq.3) to quantitatively evaluate the likelihood of NPF, and they found that NPF mostly occurred when L is smaller than 1. A similar result has been reported by Kuang et al., (2010), and a slightly different threshold L value 0.7 was determined.

$L = \frac{CS}{[H_2SO_4]} \cdot \frac{1}{\beta_{11}}$ (Eq.3)

Here, L is dimensionless parameter representing the probability that NPF will not occur, and $\beta_{11}$ is the collision rate between H$_2$SO$_4$ vapor molecules, which is characterized as $4.4 \times 10^{-10}$ cm$^3$s$^{-1}$. Our results suggest a consistent L that most (75 percentile) S-E cases happen when L is lower than 0.73 and most (75 percentile) S-NE cases are observed when L is larger than 1.54."

Points that should be clarified.
•Line 26, p. 1: "All such clusters were observed..." In the context of this sentence, this implies that all clusters from #S=3 to infinity were observed. This is obviously not what is intended.
We explicitly mention the maximum number of H2SO4 in the sentence now:
"controlled the appearance of H$_2$SO$_4$-NH$_3$ clusters (3< #S < 13): All such clusters were observed when [HOM]/[H$_2$SO$_4$] was smaller than 30."

• p. 4, line 120: "...a best instrument..." The previous sentence acknowledges that some

fragmentation occurs. It might also be mentioned that the extent of fragmentation is not quantitatively understood. Have the authors confirmed that the APi-TOF produces less fragmentation that other mass spectrometers with different interface designs? I don't think so.

The referee is right that fragmentation is very likely to happen inside the instrument, which has not been well quantified. On the other hand, as we stated in the manuscript, no ionization can be used if we aim to measure these weakly bonded clusters. Under such circumstances, the APi-TOF is very suitable for cluster measurement. As also pointed out by the other referee, we replace "a best instrument" by "a well suited instrument".

Minor editorial corrections:
The paper contains numerous minor, distracting language errors, which I illustrate with the following examples. The text should be thoroughly edited by a native English speaker.
•p. 2, line 57: should be "as ion-induced..."
Modified.

•p. 4, lines 111-112: "..often dominates the daytime spectrum in the daytime when it is abumdant,..." ???
We removed the "daytime" in the sentence.

•p. 4, line 118: replace "comparing" with "compared"
Modified.

•p. 4, line 127: delete ","
Modified.

•p. 5, line 163: delete "In specific,"
Modified.

•p. 11, Figure 3 caption: replace "unclear is IIN..." with "unclear if IIN..."
Modified.

•p. 11, line 285: delete "however" (Alternatively, it could be separated using commas, but this would make for an awkward sentence.)
Modified.

•p. 12, line 309: "this type of days..."
Modified

•p. 12, line 310: "are conduce of IIN events..."
Our original wording seems right.

•p. 12, line 315: "has not been evidenced,..."
Modified

•p. 14, Figure 5: Difficult to distinguish blue from black boxes.
We change the black box to grey in Figure 5.

•p. 14, line 349: "This indicate the..."
Modified.

•p. 15, line 371: "The abundancy and ..."
Modified.

Kuang, C., I. Riipinen, T. Yli-Juuti, M. Kulmala, A. V. McCormick and P. H. McMurry (2010).

"An improved criterion for new particle formation in diverse atmospheric environments." Atmospheric Chemistry and Physics 10: 1-12. doi: 10.5194/acp-10-1-2010.

McMurry, P. H., M. A. Fink, H. Sakurai, M. R. Stolzenburg, L. Mauldin, K. Moore, J. N. Smith, F. L. Eisele, S. Sjostedt, D. Tanner, L. G. Huey, J. B. Nowak, E. Edgerton and D. Voisin (2005). "A Criterion for New Particle Formation in the Sulfur-Rich Atlanta Atmosphere." Journal of Geophysical Research - Atmospheres 110: D22S02. doi: 10.1029/2005JD005901.

---

## Author Comment (AC2) · 2 Jul 2018

The manuscript analyzes cluster ion data from the SMEAR station in Hyytiala, Finland, from three springtime measurement periods. Data from anion measurements with an APi-TOF mass spectrometer and an NAIS instrument are analyzed. The focus of the analysis is on H2SO4-NH3 cluster ions in comparison to HOM ions and their relation to aerosol nucleation events. It is found that the ratio between [HOMs] and [H2SO4] controls the presence of large H2SO4-NH3 clusters. Furthermore, the probability for IIN to occur is largest and reaching almost 100% when clusters containing 6 or more H2SO4 molecules are present. The contribution of IIN to the total nucleation is reported to range between 4 and 45%, with an average of 12% contribution for cases that are dominated by H2SO4-NH3 nucleation and 18% in HOM-driven events. The manuscript is an extension of a series of papers focusing on results from the APi-TOF measurements in Hyytiala (e.g. Ehn et al., 2010 and 2011, Schobesberger et al., 2013 and 2015; Yan et al., 2016; Bianchi et al., 2017). The previous papers focused mostly on the role of HOMs while this one focuses on the role of H2SO4-NH3 anion clusters, therefore the paper presents sufficient new material to warrant publication in ACP. There is a number of minor points and technical corrections to consider before publication in ACP:

We would like to thank the referee for the helpful and detailed comments and suggestions.

In the following, we reply to the referee's comments item by item.

Minor points
line 43: The paper by Dunne et al., Science, 2016 should be referenced here as well.
Agreed. The paper is referenced

136: quantification of the APi-TOF results. Was the transmission of the APi-TOF characterized as described by Heinritzi et al., AMT, 2016? Can you be sure that the transmission did not change due to the changes in tuning (l 139)?
First, we believe the question is about quantification with CI-APi-TOF.
The data used here were not corrected for transmission calibration, as this instrumental fact was not recognized back to the years when the measurement was done. On the other hand, to obtain a systematic dataset of sulfuric acid and HOM concentration, the voltage tuning of the instrument was not very different between years included in this work. We attached a year-distinguished figure using the same data as in Figure 2B, in which we can see data from different years are well-mixed.

[Figure]

Figure R1. The effect of concentration of HOMs and H$_2$SO$_4$ on the appearance of H$_2$SO$_4$-NH$_3$ clusters. Symbols represent data from different years, circles for 2011, squares for 2012 and triangles for 2013.

181: What about recombination with ions larger than 3.5 nm?

In our calculation: the loss term of ions in the JIIN equation are divided into two terms:

1 – Ion-Ion recombination: the ions in the size bin (2.5-3.5 nm) become neutrals however stay in the size bin itself, since they recombine with < 3.5 nm ions ($4^{th}$ term in Eq 2).

2 – Coagulation sink: Ion loss term to coagulation which is the loss of ions to bigger sized particles or ions (>3.5 nm), leading to their loss outside of the size bin ($2^{nd}$ term in Eq 2).

Figure 2: panel B is as important as panel A. Why is B just shown as a small inset? Please show B as a separate panel of the same size as A, or even as a separate Figure.

Agreed. The Figure 2B was made in parallel with Figure 2A.

347: Please explain in detail how $J_{IIN}$ for 2.5 nm particles was calculated (here, or in Section 2).

Formation rates of 2.5 nm particles (J2.5) and ions (JIIN 2.5) are calculated using equations Eq (1) and (2) in section 2 lines 176 and 184, respectively.

404-575: Please check all references carefully: In many cases there are co-authors missing (and no "et al." is included), e.g. Bianchi et al., 2017, Dada et al., 2017, Ehn et al., 2010, Ehn et al., 2011, Kulmala et al., 2004, Schobesberger et al., 2013 and 2015, and even in Yan et al., 2016, and many others.

Modified.

General comment on choice of cited references: There is no doubt that the Kulmala group has produced lots of important research with respect to ground-based cluster ion composition measurements with the APi-TOF in Hyyttiälä, and it is therefore ok to reference the previous work of your own group frequently. Nevertheless, there have been various contributions to the field of H2SO4-NH3-IIN by other groups and the choice of references discussed for example in the introduction seems somewhat unbalanced. Out of the 34 references listed in the references section, 29 are from the Kulmala group or co-authored by the Kulmala group (and the 5 remaining references are mainly general ones such as reviews or the IPCC report). It is expected in scientific publications to give reference also to the previous work by others that is relevant for your work. Therefore I suggest to mention/discuss also work from other groups, e.g. Eisele et al., JGR, 2006; Iida et al., JGR 2006, Tammet et al., Atm. Res. 2014; Rose et al., ACP, 2013; Boulon et al., ACP 2010; Kurten et al., JGR, 2016; Froyd and Lovejoy, JPC, 2011, etc. to give some credit also to the rest of the scientific world that performed measurements of H2SO4-NH3 ion induced nucleation and other ion clusters. Also Bianchi et al., Science, 2016; Dunne et al., Science, 2016, and Wagner et al., ACP, 2018, should be included and discussed in the context of this paper (I recognize that these are also coauthored/authored by the Helsinki group).

The reviewer is right that some previous work about IIN should also be referenced. According to their relevance to this work, we have added Eisele et al., 2006; Iida et al., 2006; Kurten et al., 2016, Lovejoy et al., 2004; Wagner et al., 2017; Dunne et al., 2016 into the reference list.

Technical corrections:

line 65 and 67. Bianchi et al. is referenced twice, one time within a sentence is sufficient.modifed

Modified.

75: understandings to understanding

Modified.

92: insert space between semicolon and Ehn, as well as between semicolon and Yan

Modified.

111: "daytime spectrum in the daytime…" avoid duplication

Modified by removing "daytime"

113: "of an ion in the APi-TOF…"

Modified.

115: "note that the APi-TOF…"
Modified.

119: "instruments" to "instrument"
Modified.

120: "is a best instrument" to "is a good…" or "is a well-suited…."
We replace "a best" to "a well-suited".

127: not the same in the three years…
Modified.

129: "clusters contained 6 clusters" to "clusters contained 6 SA molecules"
Modified.

130: "in the clusters were observed" to were observed in the clusters
Modified.

131: larger than 700 Th for the measurements in 2011.
Modified.

133: Figure 3 is called here before Figure 2 is called. Change order of Figures 2 and 3.
We feel the logic flow goes better with the current figure order. Instead, we avoid calling Figure 3 by rephrasing the sentence to
"because clusters consisting of 6 $H_2SO_4$ molecules had little difference from larger clusters in affecting the IIN in terms of occurrence probability (see more details in Sect. 3.3.1)."

139: tunings of CI-APi-TOF $\rightarrow$ tuning of the CI-APi-TOF
Modified.

181, 184: "Eq. S2" $\rightarrow$ change numbering of the Equation to "Eq. 2"
Modified.

197: A similar approach
Modified.

204: measurements $\rightarrow$ measurement
Modified.

204: dependant $\rightarrow$ dependent
Modified.

242: prevents $\rightarrow$ suppresses
Modified.

261: "by observing an increase…" (delete "of")
Modified.

262: "sub-2nm ions", I think you mean "ions larger than 2 nm" here?
We meant the total signal of ions up to 2 nm. We rephrase this sentence to "… by observing an increase in the total concentration of sub-2 nm ions"

279: "…unclear is IIN occurred was counted…" → "…unclear if IIN occurred were counted…"
Modified.

289: permanence → continuity
Modified.

319 and 324: the other type of events
Modified.

321: less → lower
Modified.

322: clusters at high temperatures that can evaporate NH3 back to the atmosphere.
Modified.

340: Figure 5, panel B: "cloudiness parameter" should probably be "clear-sky parameter", or it should be explained that 1 = clear-sky = 0% cloudiness; and 0 = 100% cloudiness
We agree that the terminology is a bit confusing. For consistency, we do not change it, but instead, we add the definition in the text:
"The clear-sky parameter (100% = clear sky and 0% = cloudiness) shows a noticeably higher value during both event types compared to the non-event cases (Fig. 5B)"

349: This indicates
Modified.

361: Figure 6, panels B and C could be depicted with identical y-range (e.g. 10^-2 to 20), then a comparison would be easier. At least some tick marks should be added to panels B and C.
Modified. The color for "O-E" is changed to blue in order to be consistent with Figure 5.

368 "Summary and Conclusions" → "Summary" (there are no new conclusions, it is just a summary of the findings presented previously)
Modified.

380: from → for
Modified.

382: on other days
Modified.

385: a mechanism
Modified.

385: at least responsible for → responsible for at least
Modified.

404: Reference → References
Modified.

---

## Referee Report (RR1)

**Second review of Yan et al., acp-2018-187**

I am happy with the replies except for the reply to my comment about line 347 (line numbers with respect to the original ACPD manuscript):

First, there seem to be three different nomenclatures used in the manuscript: Are $J_{IIN}$ (line 346), $J_{2.5}^{\pm}$ (line 184 ) and $J_{Ion}$ (Figure 6, y-axis) in your definition meant to be identical? This is stated nowhere, but seems to be the case from your (very short) answer to my comment; please use identical nomenclature in all cases if it is supposed to be the same thing. Note that this nomenclature is **not** identical to the nomenclature used in Wagner et al, 2017. In Wagner et al. $J_{IIN}$ is defined as $J_{IIN} = J^{\pm} + J_{rec}$ which is correct in my opinion, as the true ion-induced nucleation rate (i.e. "particles that overcame the nucleation barrier as ions") is given by this expression. Therefore, if you want to discuss the "ion-induced nucleation rate ($J_{IIN}$) and its contribution to the total formation rate ($J_{IIN}/J_{total}$)" (l. 346-347) then you need to introduce this definition for $J_{IIN}$ and you need to introduce and calculate $J_{rec}$ and include it properly in your numbers for $J_{IIN}$ and the ion-induced fraction ($J_{IIN}/J_{total}$). Note that it is not really of interest how big the charged formation rate $J_{2.5}^{\pm}$ is for 2.5 nm particles but rather the true charged nucleation rate at the critical cluster size $J_{IIN}$.
To illustrate the important difference: Imagine a situation where all nucleation is ion-induced (like in a Wilson cloud chamber, where neutral nucleation is completely suppressed but supersaturation is sufficient for ion-induced nucleation to take place). In a situation where the subsequent growth is slow compared to the recombination (which is typically the case), all (or almost all) the charged particles that were formed by IIN originally would recombine before reaching the size of 2.5 nm. In this case $J_{2.5}^{\pm}$ would be zero or very small compared to $J_{2.5}$ and your definition returns a value for $J_{IIN}/J_{total}$ that is zero or very small, although it should be 1!
Therefore, I do not agree with your interpretation and numbers for $J_{IIN}$ and $J_{IIN}/J_{total}$ as presented in Section 3.5 and Figure 6.

Second, $J_{total}$ is also not introduced in the text, I assume it is meant to be $J_{total} = J_{2.5}^{\pm} + J_{2.5}$. (which would be identical to $J_{total} = J_{n,tot} + J_{\pm} = J_n + J_{rec} + J_{\pm}$ in Wagner et al.)

Third, I do not agree with eq. 1 in line 176. I think it needs to include a term $-\alpha N_{2.5-3.5}^{\pm} N_{<3.5}^{\mp}$ and a term $+\beta N_{2.5-3.5} N_{<2.5}^{\pm}$ to reflect the gain of neutral particles from ion-ion recombination and the loss of neutral particles due to ion-neutral collisions, just symmetric to the definition of $J^{\pm}$. Therefore, the $J_{2.5}$ data should be reanalyzed.

Fourth, please discuss the assumed value of $\beta = 1 \times 10^{-8}$ cm$^3$ s$^{-1}$ (l. 189), I don't see why this value should/could be larger than the kinetic limit for ion-molecule collisions which is around $2.4 \times 10^{-9}$ cm$^3$ s$^{-1}$ (e.g. Viggiano et al., J Phys. Chem., 1997).

Fifth, all the added references (Eisele et al., 2006; Iida et al, Lovejoy et al., 2004, Wagner et al., 2017, etc.) are not included in the list of references. Please include.

I am sorry to bring all of this up during the second round of review (and I should have noted the third and fourth point already during my first review), but this issue needs a much more thorough discussion than your 2 lines of answer to my comment about line 347.

---

## Author Response (AR2)

**Second review of Yan et al., acp-2018-187**

I am happy with the replies except for the reply to my comment about line 347 (line numbers with respect to the original ACPD manuscript):

We thank the reviewer for the valuable comments and suggestions. We provide one-to-one answers to the comments in blue below, and the respective modifications in the main text are in purple.

First, there seem to be three different nomenclatures used in the manuscript: Are JIIN (line 346), J2.5 (line 184 ) and JIon (Figure 6, y-axis) in your definition meant to be identical? This is stated nowhere, but seems to be the case from your (very short) answer to my comment; please use identical nomenclature in all cases if it is supposed to be the same thing. Note that this nomenclature is not identical to the nomenclature used in Wagner et al, 2017. In Wagner et al. JIIN is defined as JIIN = J(+/-) + Jrec which is correct in my opinion, as the true ion-induced nucleation rate (i.e. "particles that overcame the nucleation barrier as ions") is given by this expression. Therefore, if you want to discuss the "ion-induced nucleation rate (JIIN) and its contribution to the total formation rate (JIIN/Jtotal)" (l. 346-347) then you need to introduce this definition for JIIN and you need to introduce and calculate Jrec and include it properly in your numbers for JIIN and the ion-induced fraction (JIIN/Jtotal). Note that it is not really of interest how big the charged formation rate J2.5(+/-) is for 2.5 nm particles but rather the true charged nucleation rate at the critical cluster size JIIN. To illustrate the important difference: Imagine a situation where all nucleation is ion-induced (like in a Wilson cloud chamber, where neutral nucleation is completely suppressed but supersaturation is sufficient for ion-induced nucleation to take place). In a situation where the subsequent growth is slow compared to the recombination (which is typically the case), all (or almost all) the charged particles that were formed by IIN originally would recombine before reaching the size of 2.5 nm. In this case J2.5(+/-) would be zero or very small compared to J2.5 and your definition returns a value for JIIN/Jtotal that is zero or very small, although it should be 1! Therefore, I do not agree with your interpretation and numbers for JIIN and JIIN/Jtotal as presented in Section 3.5 and Figure 6.

We appreciate the reviewer's point as it will drastically improve the consistency of the paper and prevent confusions towards the nomenclature used throughout the text. We also apologize for not addressing the problem properly during the first round of review. Although we agree with the calculation presented in Wagner et al. (2017), we must base our analysis on the available atmospheric measurements and build on available atmospheric literature. Anyway, we made substantial revisions to the text, as detailed below:

First, we made it clearer what we mean by the formation rates of ions and particles obtained from our measurements (equations 1 and 2, see our response to the comments 2 and 3 below).

Second, we rewrote the beginning of section 3.5 to address the main concerns pointed out by the reviewer. It now reads:

"In order to get further insight into the importance of IIN during our measurements, we compared the formation rate of 2.5 nm ions, $J_{ION} = J_{2.5}^{\pm}$ (see Eq.2) to the total formation rate of 2.5 nm particles, $J_{TOT} = J_{2.5}$ (see Eq.1). The ratio $J_{ION}/J_{TOT}$ is equal to the charged fraction of the 2.5 nm particle formation rate. In analyzing field

measurements, a similar ratio at a certain particle size (typically 2 nm) has commonly been used to estimate the contribution of ion-induced nucleation to the total nucleation rate (see Hirsikko et al. 2011, and references therein). It should be noted that $J_{ION}/J_{TOT}$ represents only a lower limit for the contribution of ion-induced nucleation, as this ratio does not take into account the potential neutralization of growing charged sub-2.5 nm particles by ion-ion recombination (e.g. Kontkanen et al., 2013; Wagner et al., 2017). At present, measuring the true contribution of ion-induced nucleation to the total nucleation rate is possible only in the CLOUD chamber (Wagner et al., 2017). We were able to calculate $J_{ION}$ and $J_{TOT}$ for 57 (out of 67) cases, and the ratio $J_{ION}/J_{TOT}$ varied from 4 to 45%, showing a clear correlation with the HOM signal (Fig. 6A)…"

Third, we rewrote the caption of Figure 6 into the following form:

"A) Charged fraction of the formation rate of 2.5 nm particles as a function of the total signal of HOM ions color-coded by the $H_2SO_4$ concentration, and (B, C and D) the differences in $J_{ION}$, $J_{TOT}$, and $J_{ION}/J_{TOT}$ between the $H_2SO_4$ – $NH_3$-involved events (S-E) and other events (O-E)."

Finally, we removed the text ", contributing up to 40% of the total nucleation rate" from the last paragraph of section 4.

Hirsikko, A., Nieminen, T., Gagne, S., Lehtipalo, K., Manninen, H. E., Ehn, M., Horrak, U., Kerminen, V.-M., Laakso, L., McMurry, P. H., Mirme, A., Mirme, S., Petäjä, T., Tammet, H., Vakkari, V., Vana, M., and Kulmala M.: Atmospheric ions and nucleation: a review of observations, Atmos. Chem. Phys., 11, 767-798, 2011.

Kontkanen, J., Lehtinen, K. E. J., Nieminen, T., Manninen, H. E., Lehtipalo, K., Kerminen, V.-M., and Kulmala, M.: Estimating the contribution of ion-ion recombination to sub-2 nm cluster concentrations from atmospheric measurements, Atmos. Chem. Phys., 13, 11391-11401, 2013.

Second, Jtotal is also not introduced in the text, I assume it is meant to be Jtotal = J2.5$^{\pm}$ + J2.5. (which would be identical to Jtotal = Jn,tot + J± = Jn + Jrec + J± in Wagner et al.)

Third, I do not agree with eq. 1 in line 176. I think it needs to include a term to reflect the gain of neutral particles from ion-ion recombination and the loss of neutral particles due to ion-neutral collisions, just symmetric to the definition of J. Therefore, the J2.5 data should be reanalyzed.

We thank the reviewer for these two comments. However, there appears to be a misunderstanding, probably because we were a bit unclear in defining the quantities in equation 1. The quantity $J_{2.5}$ represents the total formation rate of 2.5 nm particles obtained from measurements, not just the neutral fraction of these particles (with some influence by recombination products). As a result, there is no need for equations 1 and 2 to be symmetric with respect to the terms representing ion-ion recombination or ion-aerosol attachment.

In order to avoid the potential confusion noted by the reviewer, we modified the sentence prior to equation 1 into the form: "The formation rate of 2.5 nm particles includes both neutral and charged particles, and it was calculated from the following equation:"

Furthermore, the sentence prior to equation 2 as modified as: "Calculating the formation rate of 2.5 nm ions, or charged particles, includes two additional terms....".

Fourth, please discuss the assumed value of B = 1 x 10-8 cm3 s-1 (l. 189), I don't see why this value should/could be larger than the kinetic limit for ion-molecule collisions which is around 2.4 x 10-9 cm3 s-1 (e.g. Viggiano et al., J Phys. Chem., 1997).

We thank the reviewer for his comment; however we do not see how 2.5 nm ion-neutral collision rate would be related to molecule – sulfuric acid reaction rates presented in the referenced article (e.g. Viggiano et al., J Phys. Chem., 1997) entitled Rate Constants for the Reactions of XO3 -(H2O)n (X ) C, HC, and N) and NO3 -(HNO3)n with H2SO4: Implications for Atmospheric Detection of H2SO4. Anyway, we added the following text to the end of the paragraph following equation 2:

"We consider these values as reasonable approximations, keeping in mind that the exact values of both $\alpha$ and $\beta$ depend on a number of variables, including the ambient temperature, pressure and relative humidity as well as the sizes of the colliding objects (ion-ion or ion-aerosol particle) (e.g. Hoppel, 1985; Tammet and Kulmala, 2005; Franchin et al., 2015).

Hoppel, W. A.: Ion-aerosol attachment coefficients, ion depletion, and the charge distribution on aerosols, J. Geophys. Res., 90, 5917-5923, 1985.

Franchin, A., Ehrbart, S., Leppä, J., Nieminen, T., Gagne, S., Schobesberger, S., Wimmer, D., Duplissy, J., Riccobono, F., Dunne, E. M., Rondo, L., Downard, A., Bianchi, F., Kupc, A., Tsagkogeorgas, G., Lehtipalo, K., Manninen, H. E., Almeida, J., Amorim, A., Wagner, P. E., Hansel, A., Kirkby, J., Kurten, A., Donahue, N. M., Makhmutov, V., Mathot, S., Metzger, A., Petäjä, T., Schnitzhofer, R., Sipilä, M., Stozhkov, Y., Tome, A., Kerminen, V.-M., Carslaw, K., Curtius, J., Baltensperger, U., and Kulmala, M.: Experimental investigation of ion-ion recombination under atmospheric conditions, Atmos. Chem. Phys., 15, 7203-7216, 2015.

Fifth, all the added references (Eisele et al., 2006; Iida et al, Lovejoy et al., 2004, Wagner et al., 2017, etc.) are not included in the list of references. Please include.

Thank you for pointing this out, we updated our reference list.

I am sorry to bring all of this up during the second round of review (and I should have noted the third and fourth point already during my first review), but this issue needs a much more thorough discussion than your 2 lines of answer to my comment about line 347.